# Control Effects of Hydraulic Interception Wells on Groundwater Pollutant Transport in the Dawu Water Source Area

**Henghua Zhu [1,2], Jianwei Zhou [1,\*], Chao Jia [3,\*]** **, Sheng Yang [3], Jing Wu [3], Lizhi Yang [2], Zhengrun Wei [2], Hongwei Liu [4] and Zhizheng Liu [2]**

1   School of Environmental Studies, China University of Geosciences, Wuhan 430074, China
2   Shandong Institute of Geological Survey, Jinan 250013, China
3   Institute of Marine Science and Technology, Shandong University, Qingdao 266237, China
4   Tianjin Center, China Geological Survey, Tianjin 300170, China
\*   Correspondence: jw.zhou@cug.edu.cn (J.Z.); jiachao@sdu.edu.cn (C.J.);
    Tel.:+86-135-0719-6575 (J.Z.); +86-135-0531-3402 (C.J.)

**Abstract:** Based on the comprehensive analysis of hydrogeological data of the Dawu water source in Zibo city, the Modflow module in Groundwater Modeling System is used to carry out three-dimensional geological modeling of the Dawu water source, and the flow field model and solute transport model of the Dawu water source are established. Aiming at the problem of groundwater pollution in the key polluted area of the Hougao region—the Dawu water source—the pollutant transport model is established to explore the process of pollution transport. There are many types of groundwater pollutants in the Hougao area. Among them, ammonia nitrogen, chloride, petroleum, and benzene exceed the standard most seriously. In order to facilitate the research, we selected typical pollutants for in-depth study. The ammonia nitrogen is used as the control index of domestic and industrial waste water in the policy documents of pollution emission. It can show the specific situation of industrial waste water and domestic waste water pollution changing with time. Thus, the ammonia nitrogen with a higher exceeding standard is selected as the pollution factor in this simulation. Pollutant transport under the conditions of strong pumping and stop pumping is simulated. It is found that the pollutant is effectively controlled due to the pumping and discharging effects under the action of strong pumping, from 4 to 5 times exceeding the standard to slightly exceeding the standard. However, there is still a trend of migration to the eastern water supply area. After the pumping is stopped, the pollutant quickly migrates to the Xixia centralized water supply area, causing serious pollution to the water supply area. Finally, four other hydraulic interception wells are set up in the 500 m east of Hougao's four wells to further control the pollutant transport. When hydraulic interception wells and strong pumping wells are used together, the scope of ammonia nitrogen pollution is basically controlled near the interception wells, and it does not continue to spread to the eastern water supply area. The maximum monitoring value of pollution is 0.11 mg/L, which is controlled within the standard limit of three types of groundwater, and the pollutant control effect is the best, providing certain reference for similar pollution control work.

**Keywords:** Dawu water source area; groundwater pollution; numerical simulation; pollution control

## 1. Introduction

The Dawu water source area is a rare super-large underground water source area in Northern China, and it is the main source of urban water supply to Zibo city, which has important strategic significance for a water resource guarantee and water supply security for Zibo city [1]. The specific

scope of water source includes west of Zi river, east of Fengbei road, north of Liuzheng village and Xuwang village, and the closed area south of national highway 309. The total area of the water source is about 148 km$^2$. Figure 1 shows the location of Dawu water source area.

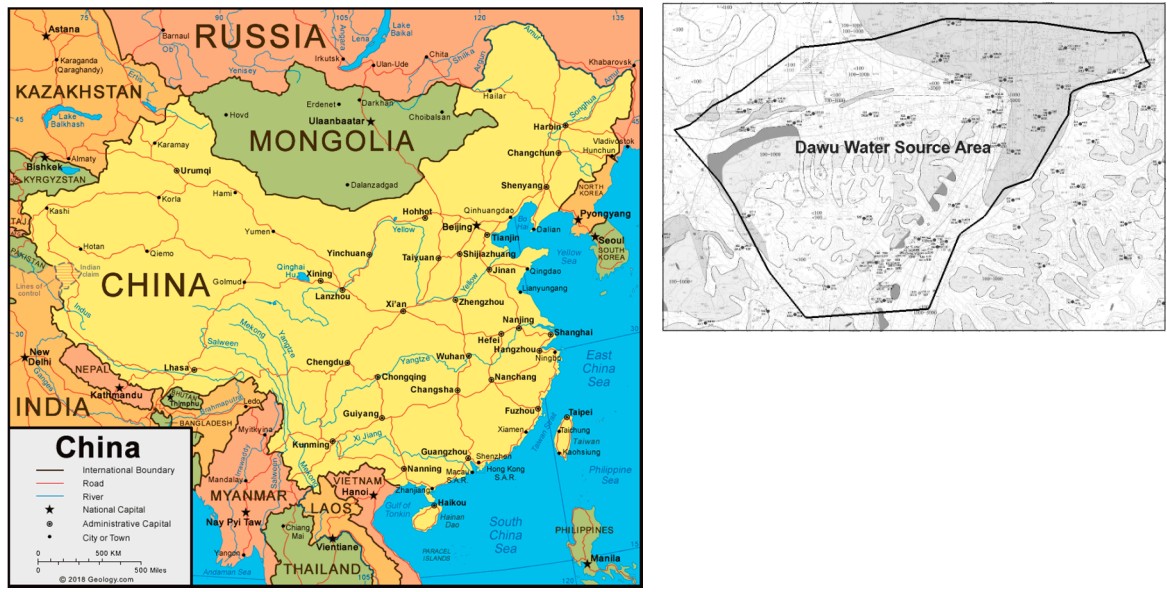

**Figure 1.** The location of the Dawu water source area.

The Dawu water source area is a complete hydrogeological unit, in which the water-bearing rocks are mainly the pore water-bearing rocks of the upper quaternary loose rocks and the fissure karst water-bearing rocks of the lower Ordovician carbonate rocks. Pore water supply is mainly atmospheric precipitation infiltration, and water quality is relatively poor and generally used for agricultural irrigation. The recharge source of karst water is mainly the infiltration of atmospheric precipitation in the southern mountains and the leakage of the Zi river. The water quality is relatively good, and the water is very rich. It is mainly used for residential, urban, and industrial water, and it is the main mining layer of the Dawu water source. The water source has a strong capacity of water storage and regulation, among which there are about 300 deep wells. The water inflow of a single well is generally about 200 m$^3$/h and the maximum is about 450 m$^3$/h. Since the Yellow River diversion water supply project of Zibo city was put into operation in 2001, the exploitation volume remained at about 360,000 m$^3$/d, and the annual and inter-annual variation of groundwater level was large, with the maximum annual variation range of 50–60 m. Since the exploitation of the Dawu water source in 1960, it has provided a strong water supply guarantee for Zibo urban residents' living water supply and production water supply for large state-owned enterprises, such as Sinopec Qilu Petrochemical and so on in the Linzi area. Because of its strong capacity of water storage and regulation, the Dawu water source area is of great strategic significance to the water resource security and water supply safety of Zibo city.

In recent years, the Dawu water source area has adhered to the principles of ecological civilization, environmental coordination, and sustainable development. However, due to historical reasons, since the mid-1960s, Sinopec Qilu Petrochemical company's Shengli oil refinery, rubber plant, first fertilizer plant, second fertilizer plant, and other large chemical enterprises have been built in the limestone area in the east of the water source area. In the 1980s, large ethylene plants, such as olefin plants, chlor-alkali plants, and plastic plants, were built successively in the limestone areas in the west of the water source area. The distribution of these chemical enterprise installations is basically the same as that of the rich underground water area in the water source area. The surface water environment of the water source area is extremely severe, and the ground pollution or quaternary soil pollution is widely distributed. Multiple pollution areas are formed in the water source area, resulting in different

degrees of pollution of multiple wells in the water source area [2,3]. Especially, the pollution in Hougao area with more chemical plants is the most serious. According to the monitoring data of the water quality monitoring points in the Dawu water source area in recent years which is shown in Tables 1 and 2, the pollutant concentration exceeding the limit is always the most serious in four wells in the Hougao area, among which the chloride reaches 1140 mg/L at the highest point, 4.56 times higher than the IIIstandard. Ammonia nitrogen reached 151 mg/L at the highest point, 302 times higher than the IIIstandard. The highest level for petroleum was 5.5 mg/L, 275 times higher than the IIIstandard. Benzene reached 58.2 mg/L at the highest value, 5820 times higher than the IIIstandard. It shows that the Hougao region is seriously polluted and it is the key pollution area of the Dawu water source. Even though the Hougao area has adopted strong mining measures to reduce the concentration of pollutants in groundwater since the 1990s, the concentration of pollutants in the Hougao area still exceed the standard limit of groundwater quality several times from the recent monitoring data from March 2016 to May 2017.

**Table 1.** Water quality monitoring data of Hougao's four wells in March 2016 (mg/L).

| Well | The Well Number | Chloride | Ammonia Nitrogen | Petroleum | Benzene |
|---|---|---|---|---|---|
| Hougao1# | HG1 | 581 | 10.7 | 3.32 | 0.113 |
| Hougao2# | HG2 | 603 | 7.7 | 2.68 | 0.977 |
| Hougao3# | HG3 | 383 | 8.95 | 5.5 | 12.9 |
| Hougao4# | HG4 | 484 | 8.84 | 1.63 | 0.141 |

**Table 2.** Water quality monitoring data of Hougao's four wells in May 2017 (mg/L).

| Well | The Well Number | Chloride | Ammonia Nitrogen | Petroleum | Benzene |
|---|---|---|---|---|---|
| Hougao1# | HG1 | 550.6 | 10.02 | 1.12 | 0.086 |
| Hougao2# | HG2 | 564.4 | 11.38 | 0.928 | 1.31 |
| Hougao3# | HG3 | 261.6 | 13.13 | 12.02 | 17.1 |
| Hougao4# | HG4 | 366.5 | 6.4 | 1.235 | 0.126 |

At present, analytical and numerical methods are the main methods used in the study of pollutant transport [1–5]. The numerical method is effective for the simulation of pollutant transport in complex geological conditions. The most commonly used numerical simulation software is Modflow, developed by the U.S. Geological Survey in the 1980s. At present, there are a large number of successful cases of pollutant transport using Modflow at home and abroad. For example, Shen et al. [6] used this software to simulate the movement of groundwater pollution in a chemical plant in Shanghai. Xu et al. [5] used Modflow to simulate groundwater pollution in karst areas in southwest China. Li et al. [7] used this software to simulate and predict groundwater pollution in typical areas of the Hunhe fan. Seyed et al. [8] used this software to simulate the pollutant migration in confined water aquifers. Therefore, the four Hougao wells as the key pollution source are taken as the research background, and the temporal and spatial migration of ammonia nitrogen under the action of strong pumping and stop pumping is simulated using the groundwater simulation software Modflow. At the same time, the blocking effects of the other four hydraulic interception wells on the pollutant under the action of strong discharging are simulated. The research results can provide a reference for the treatment of similar polluted sites.

## 2. Establishment of Hydrogeological Model

### 2.1. Research Scope and Determination of Boundary Conditions

The Dawu water source area is located in the mid-east region of Zibo city, mainly in the Linzi district of Zibo city. The west includes the partial region of the Zhangdian district, Fengshui town. The south includes the partial villages and towns northeast of the Zichuan district. The east involves the

western region of Weifang city and Qingzhou city. Geographical coordinates are as follows: northern latitude 36°30′31″–36°51′12″, and east longitude 118°02′16″–118°26′55″. The key simulated area is the Qilu Petrochemical sewage treatment plant, which is located in the west of the Dawu water source area and the east of the Jinling fault. For the convenience of research, the sewage treatment plant is taken as the core and a certain distance is extended to the periphery as the range of the model. The south and north sides are basically perpendicular to the groundwater flow line, as the zero flow boundary, and the east and west sides are parallel to the flow line, as the constant flow boundary. The boundary flow is calculated by the cross section method.

*2.2. Generalization of Source and Sink Terms*

Rainfall evaporation: The climate of the working area (Linzi district) has a warm temperate continental monsoon climate. The annual average temperature is 12.2 °C. The average precipitation is 648.4 mm. The maximum annual precipitation was 1118.7 mm in 1964, and the minimum annual precipitation was 319.5 mm in 2006. The annual distribution of precipitation is uneven, and the annual variation is great. The area is located in a hilly area. The distribution of precipitation in the region is unevenly affected by geographical location, terrain, and other factors. The trend of the precipitation contour line and the topographic contour line is roughly the same, basically showing an east-west trend. The distribution trend of annual average precipitation is decreasing from south to north. Due to the influence of local terrain, the precipitation center is in the area of the Heiwang iron mine and Miaozi district. The annual change process of precipitation presents the alternation of abundant, and flat and dry. Precipitation in dry years is only about 60% of the annual average. The rainfall infiltration coefficient is the ratio of precipitation infiltration supply in a certain period to the corresponding precipitation in the same period. Additionally, it is related to many factors, such as groundwater depth, early soil moisture content, lithology, and vegetation. The precipitation infiltration recharge coefficient correlates with the groundwater depth. According to the engineering exploration data and technical requirements of hydrogeological parameter acquisition methods, the rainfall infiltration recharge coefficient is set at 0.15, and it is assigned with a recharge module.

The annual average evaporation is 1500–1900 mm. The distribution law is basically the same with the precipitation. The evaporation from April to July is the largest, and accounts for more than 53% of the annual evaporation, and the monthly average evaporation exceeds 200 mm. The evaporation in winter is the smallest, and accounts for less than 10% of the annual evaporation, and the monthly average evaporation is about 50 mm. The groundwater depth of the water source is generally deep, and the average depth is 50–100 m. According to the technical requirements of hydrogeological parameter acquisition methods, the evaporation coefficient is set at 0.01, and it is assigned by the evaporation module.

Mining situation in the study area: The northeast area is mainly the water source of the Xindian power plant with 16 mining wells. The well depth is about 350–400 m, and the production capacity is about 20,000–30,000 m$^3$/d. It is the domestic and production water of the power supply plant. In addition, there are four strong drainage wells with depths of 350–400 m in the Hougao area. The producing capacity is 15,000–20,000 m$^3$/d. The groundwater enters the water treatment center and is reused after treatment. There are also a large number of company-owned wells in the region, and the producing capacity is about 50,000–100,000 m$^3$/d. The pumping value is assigned with the well module.

*2.3. Generalization of Stratigraphic Structure*

The water source area is high in the south and low in the north, with low hills in the south and piedmont sloping plain in the north. The east of the Zi river is a low and middle mountain area composed of Maan mountain, Qingliang mountain, and Ma mountain. The altitude is about 300–780 m. The west of Zi river is the surface watershed composed of mountains. The area from the south to the north gradually lowers into the piedmont sloping plain. The altitude is about 50–150 m. The terrain

elevation near the watershed is 200–700 m, which is a hilly area. The mountains are monoclinic, which are gentle in the northwest slope and steep in the southeast slope.

The working area is located in the northeast of Zibo basin, which belongs to the transition zone between the mountain area of Luzhong and the North China Plain. The south of the working area is the low hill to low hilly area, and intermontane basin and valley terrain are distributed in local areas. The north area is the north-sloping piedmont plain. According to the characteristics of aquifer medium in the Dawu water source area, the simulation is divided into two water-bearing rock groups. They are aquifer formation and artesian aquifer formation, respectively. The aquifer formation is mainly composed of quaternary unconsolidated rock pore water aquifer formation and carbonate rock fissure karst water-bearing formation. The pore aquifer formation of the quaternary unconsolidated rocks is mainly distributed in the northern part of the study area. Its thickness increases gradually from south to north, commonly 50–100 m, and the maximum thickness can get to 200 m. The groundwater quality of this aquifer formation is poor, which is mainly used for agricultural irrigation water and domestic water of a small number of rural areas. Additionally, this aquifer formation is also part of the main groundwater recharge sources of the underlying carbonate fissure karst aquifer formation.

The fractured karst aquifer formation of carbonate rocks is widespread in the study area. The lithology of its water-bearing section is argillaceous dolomite, brecciated argillaceous limestone, medium-thick layer gray limestone, and leopard skin limestone. Limestone fissure karst is relatively developed. Its depth is between 60 and 300 m with high water content. It is the largest water-rich area in the water source area and the main water-intake aquifer in the Dawu water source area.

## 3. Establishment and Identification of the Unsteady Groundwater Flow Model

### 3.1. Mathematical Model

According to the hydrogeological conceptual model of the study area, the groundwater seepage in the water source conforms to Darcy's law. The mathematical model of unsteady groundwater flow in a mining area can be established by combining continuity equations [9,10].

$$
\begin{cases}
\frac{\partial}{\partial x}\left(K_{xx}\frac{\partial H}{\partial x}\right) + \frac{\partial}{\partial y}\left(K_{yy}\frac{\partial H}{\partial y}\right) + \frac{\partial}{\partial z}\left(K_{zz}\frac{\partial H}{\partial z}\right) + \\
W = S_s\frac{\partial H}{\partial t} \qquad (x,y,z) \in \Omega, t \geq 0 \\
H(x,y,z,0) = H_0(x,y,z) \qquad (x,y,z) \in \Omega \\
-K\frac{\partial H}{\partial n}\big| = q \qquad t \geq 0, (x,y) \in B_{east,west} \\
-K\frac{\partial H}{\partial n}\big|_{B_{south,north}} = 0 \qquad t \geq 0 \\
\frac{\partial H}{\partial n}\big|_{B_{botton}} = 0 \qquad t \geq 0
\end{cases}
\tag{1}
$$

where $H$ is the head of water, m; $K_{xx}$, $K_{yy}$, and $K_{zz}$ are permeability coefficients in the x, y, and z directions, m/d; $n$ is the normal vector; $S_s$ is elastic water release rate, 1/d; $W$ is rainfall infiltration recharge intensity and evapotranspiration intensity, m$^2$/d; $H_0$ is the initial flow field in the simulation region, m; $q$ is the fixed flow boundary flow of the second type, m$^3$/d, the inflow is positive and the outflow is negative; $\Omega$ is simulation area.

### 3.2. Model Establishment and Parameter Partitioning

According to the above hydrogeological model, the simulation range, boundary conditions, source and sink terms, and other conditions were determined. The numerical model of groundwater flow field was established using the Modfolw conceptual modeling method. The model was divided into 28,303 effective grids, as shown in Figure 2.

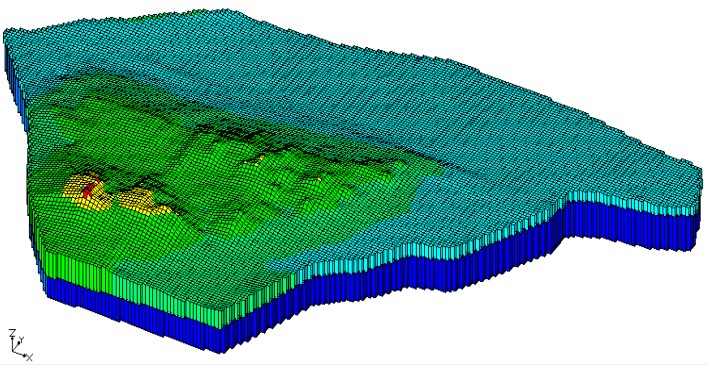

**Figure 2.** Three-dimensional structure of the hydrogeological model.

According to the hydrogeological data of the study area, the simulation was divided into two water-bearing rock formations, i.e., the water-bearing rock formation with diving water and water-bearing rock formation with confined water [11–16]. The parameter partition of each layer was generalized as shown in Figures 3 and 4. The initial value of hydrogeological parameters of each zone was preliminarily given by borehole pumping tests and other data. In the later stage, the final correction value was determined by checking and adjusting the measured groundwater level of each well point.

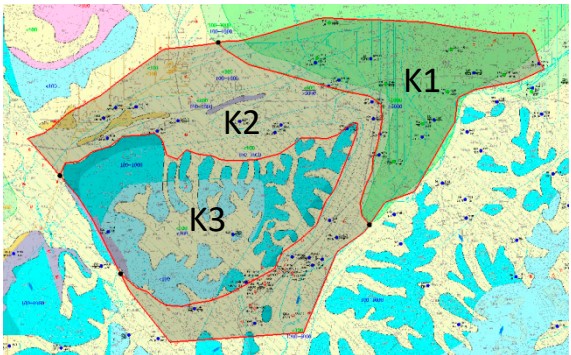

**Figure 3.** Parameter zoning map of the phreatic aquifer.

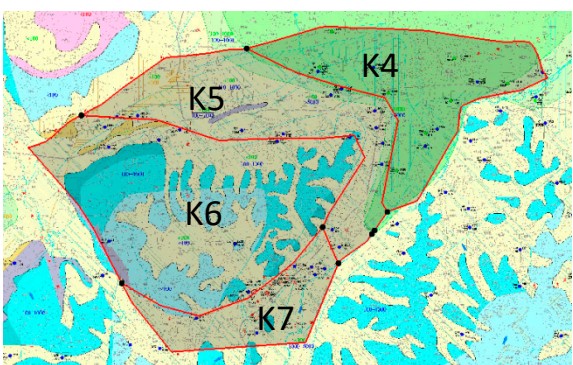

**Figure 4.** Parameter zoning map of the confined aquifer.

### 3.3. Identification and Verification of Numerical Models

According to the geological exploration data and pumping test data of the Dawu water source area, the lithology of the quaternary loose rock pore aquifer group in the Dawu water source area is medium coarse sand–sand gravel, and the permeability coefficient of the groundwater environment is about $2.89 \times 10^{-2}$–$5.78 \times 10^{-2}$ cm/s, that is, 25–50 m/d. The carbonate fissure karst water-bearing rock formation is

mainly lithologically composed of limestone, argillaceous dolomite limestone, and dolomite. It is the water supply aquifer of the Dawu water source, and it has relatively high outcrops in the mountainous areas in the east and south of China. In the north of the Dawu water source, the aquifer is mainly concealed in the quaternary system or buried under the carboniferous Permian strata, and karsts are relatively developed. A large number of dissolved pores can be seen in the borehole core, and the water is rich. With reference to the technical requirements for numerical simulation of groundwater flow formulated by the China geological survey for the investigation and evaluation of groundwater resources and environmental problems nationwide, the permeability coefficient of fractured aquifer of carbonate rock in the exposed area was set as 110 m/d–140 m/d and 50 m/d–90 m/d in the concealed area as the initial value of model fitting.

The model identification and verification are important means to check whether the model conforms to the actual hydrogeological conditions. Two independent hydrological years shall be selected for identification and inspection, respectively. The actual measured data of groundwater levels in the research area from January to December in 2016 are selected to identify the model. The model was tested with the measured groundwater level data from January to December in 2017. The calculated flow field and the measured flow field can be basically fitted by correcting the hydrogeological parameters in the study area. The hydrogeological parameters were obtained when the flow field was finally identified and adjusted, which are shown in Table 3.

**Table 3.** Hydrogeological parameters of the model.

| Aqueous Petrofabric | Partition Number | The Permeability Coefficient K (m·d$^{-1}$) | | Specific Yield | Elastic Water Release Rate |
|---|---|---|---|---|---|
| Aquifer type | | $K_x = K_y$ | $K_z$ | $\mu$ | $S_s$ (m$^{-1}$) |
| Phreatic aquifer | K1 | 50.3 | 16.6 | 0.01 | |
| | K2 | 35.8 | 8.9 | 0.007 | |
| | K3 | 70.4 | 17.6 | 0.03 | |
| | K4 | 55.6 | 18.5 | 0.01 | |
| Confined aquifer | K5 | 85.9 | 21.5 | | 0.015 |
| | K6 | 110.7 | 36.9 | | 0.015 |
| | K7 | 65.4 | 16.4 | | 0.013 |

Finally, the identified parameters, determined boundary conditions, source and sink terms, and other conditions were substituted into the model to calculate the flow field of the phreatic aquifer and confined aquifer. The calculated flow field shown in Figure 5 was well matched with the measured flow field shown in Figure 6 in streamline shape and numerical value, and the calculated flow field can basically be matched with the actual flow field in macroscopic. At the same time, Hougao no. 4 well, Xixia no. 1 well, and Xindian no.5 well were selected to carry out dynamic fitting tests for the simulated water level in the research area. The results shown in Figures 7–9 prove that the dynamic water level fitting is good, and the maximum difference is 1.503 m, which can meet the accuracy requirements of simulation.

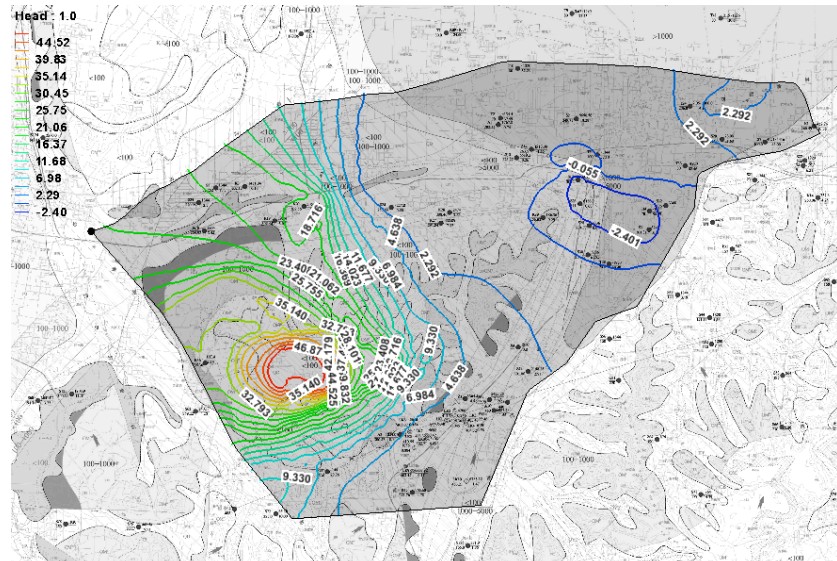

**Figure 5.** Computational flow field diagram of the research area in 2017.

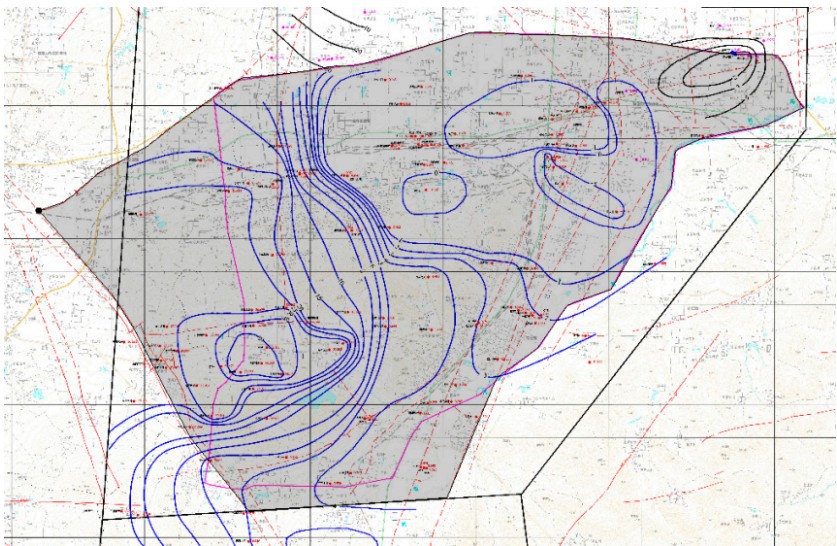

**Figure 6.** Measured flow field map of the research area in 2017.

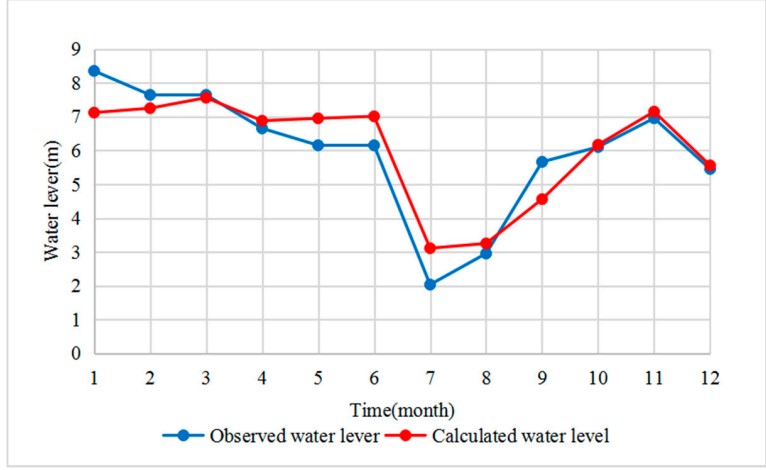

**Figure 7.** Dynamic fitting curve of groundwater level of Hougao well 4 in 2017.

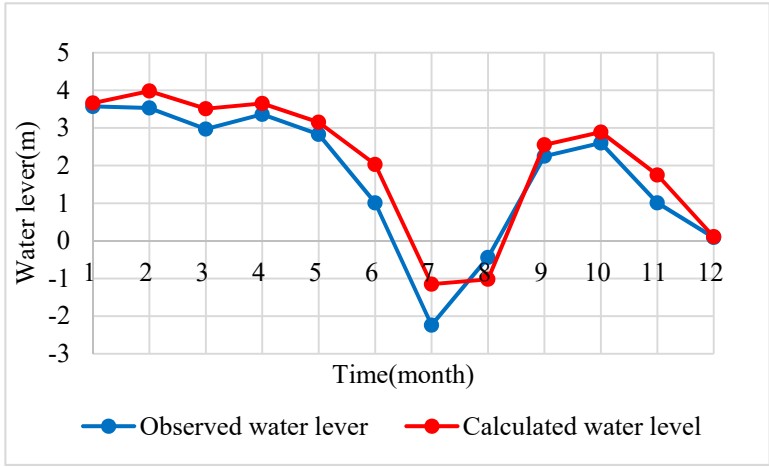

**Figure 8.** Dynamic fitting curve of groundwater level of Xixia well 1 in 2017.

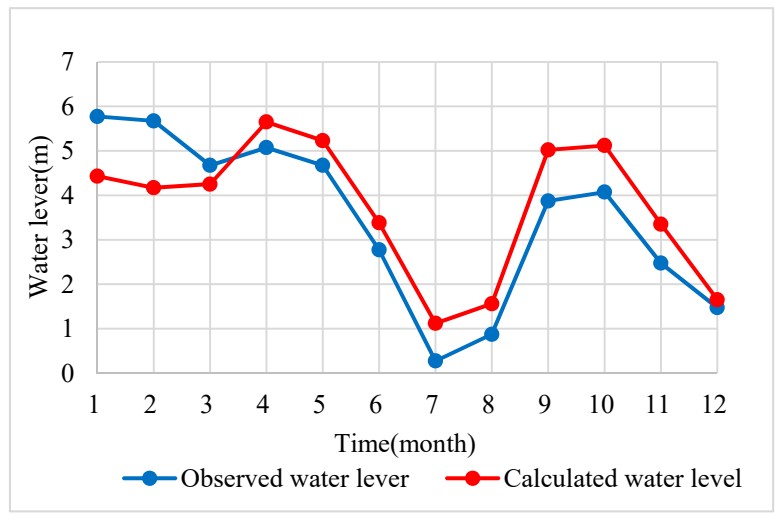

**Figure 9.** Dynamic fitting curve of groundwater level of Xindian well 5 in 2017.

## 4. Pollutant Transport Simulation and Prediction

### 4.1. Pollutant Release Scenario Setting

There are many types of groundwater pollutants in the Hougao area. Among them, ammonia nitrogen, chloride, petroleum, and benzene exceed the standard most seriously. The ammonia nitrogen is used as the control index of domestic and industrial waste water in the policy documents of pollution emission. It can show the specific situation of industrial waste water and domestic waste water pollution changing with time. Thus, ammonia nitrogen was selected as the pollution factor in this simulation. According to the latest detected concentration of ammonia nitrogen pollutants of the four Hougao wells in May 2017, the concentration of ammonia nitrogen was 10.02 mg/L in Hougao well 1#, 11.38 mg/L in Hougao well 2#, 13.13 mg/L in Hougao well 3#, and 6.4 mg/L in Hougao well 4#, which were used as the initial concentrations of the ammonia pollution simulation. According to the class III standard of the groundwater environmental quality standard of GB/T 14848-2017, the standard limit concentration of ammonia nitrogen is 0.5 mg/L. The maximum influence range of the standard limit concentration is taken as the pollution exceeding limit range [12–15]. Since the pollutant concentration of the four Hougao wells is high all year round, the pollutant release condition is set as the continuous release condition.

### 4.2. Prediction and Analysis of Pollutant Transport without Pumping

Since the 1980s, the four Hougao wells have taken strong drainage measures to control the pollutant transport from the Hougao area to the concentrated water supply area of the eastern Dawu water source. In this study, the pollutant transport of the four Hougao wells was studied when pumping was stopped. The results are shown in Figure 10. After the pumping was stopped, the ammonia nitrogen rapidly moves to the east central water supply area under the convection action of karst fissure water without taking control measures. After 20 years, the maximum influence distance of the concentration boundary corresponding to water standard limit of class III is about 4.7 km. The pollutant affects the city water department and power plant area, and the impact distance is far, which seriously affects the safety of the domestic water supply.

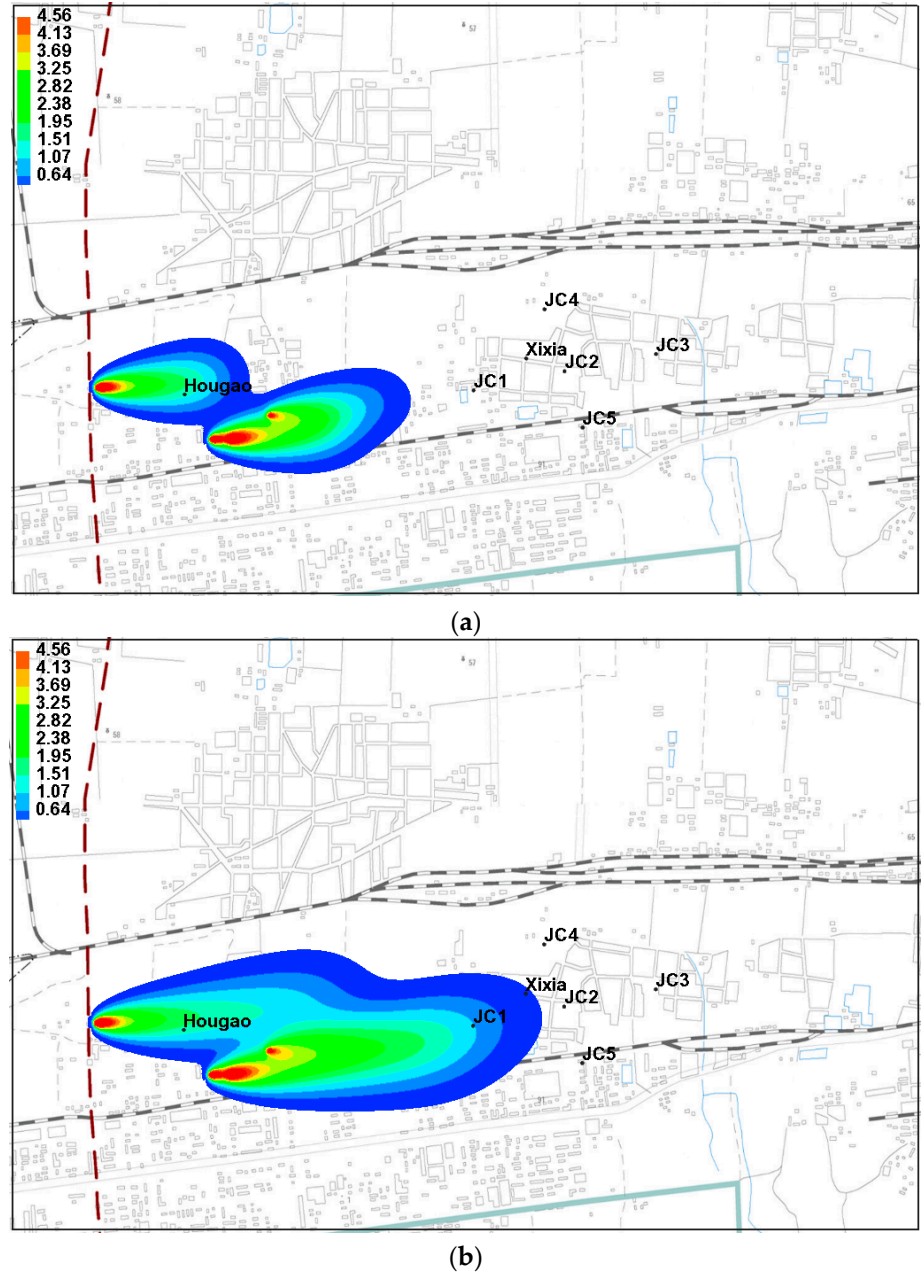

(a)

(b)

**Figure 10.** *Cont.*

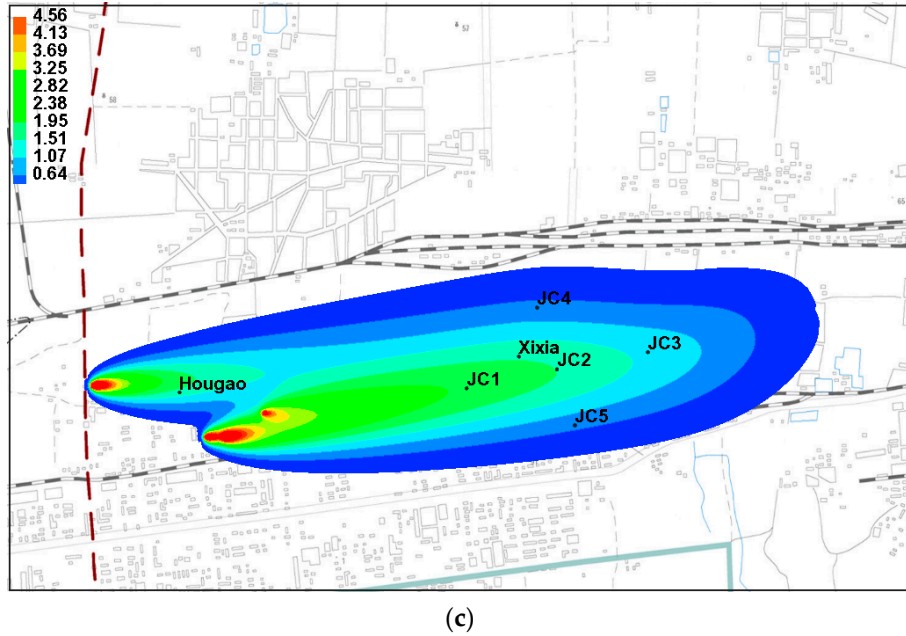

(**c**)

**Figure 10.** Prediction diagram of ammonia nitrogen transport in the four Hougao wells after pumping is stopped; (**a**) five years; (**b**) ten years; (**c**) twenty years.

At the same time, five monitoring wells (JC1–JC5) in the Xixiazhuang area were selected to observe the difference of spatial distribution of pollutants. The concentration changing curve is drawn according to the concentration monitoring data of the monitoring well, as shown in Figure 11. It can be seen from the change curve of ammonia nitrogen concentration in the monitoring wells that the ammonia nitrogen concentration in each monitoring well presents a trend from rising to stable. In the process of ammonia nitrogen migration, the concentration of ammonia nitrogen in the monitoring wells will rise due to the concentration of ammonia nitrogen in the monitoring wells. With the further expansion of the ammonia nitrogen diffusion range, the concentration of ammonia nitrogen in the monitoring wells will gradually stabilize. At the same time, by comparing the concentration curves of each monitoring well, it can be found that the closer the monitoring well to four Hougao wells source pollution, the faster the ammonia nitrogen concentration reaches the peak. Monitoring well JC1 reaches the peak of 2.32 mg/L in 29 years; JC2 2.05 mg/L in 32 years; and JC3 reaches the peak of 1.85 mg/L in 35 years. By comparing the peak concentration of monitoring wells, it can be found that the closer the monitoring wells to the pollution source, the higher the peak concentration. By comparing the concentration curves of JC2, JC4, and JC5, it can be found that the closer the area to the pollution halo center, the higher the pollutant concentration. The pollutant concentration decreases from pollution halo center to both sides. From the concentration curve, it can be seen that the concentration of ammonia nitrogen pollutants in the Xixia area is relatively high, reaching a maximum of 2.32 mg/L. The concentration is 4–5 times above class III water standard. The ammonia nitrogen concentration of JC4 and JC5 monitoring wells with smaller concentration also reach 0.84 mg/L and 0.74 mg/L, and all exceed the 0.5 mg/L stipulated in the class III of the groundwater standard. Therefore, in the absence of preventive and control measures, ammonia nitrogen pollutants will continue to maintain a relatively high concentration, endangering the water security in the eastern region.

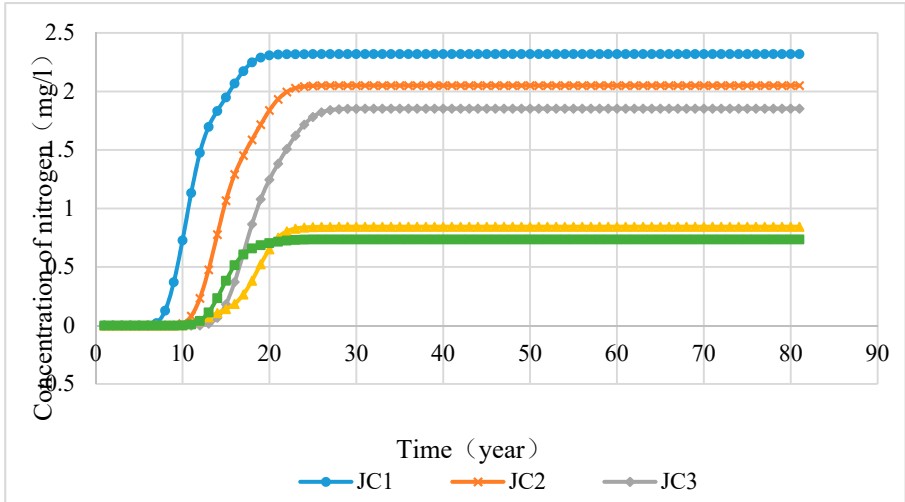

**Figure 11.** Ammonia nitrogen concentration changing curves of five monitoring wells without pumping.

### 4.3. Prediction and Analysis of Pollutant Transport under the Effect of Strong Pumping

The pollutant transport in the four Hougao wells with strong pumping is shown in Figure 12. The diffusion of ammonia nitrogen pollutant was controlled obviously. The pollution halo of high concentration was basically controlled in the vicinity of the strong pumping well, and does not continue to spread outwards. However, the super standard contaminant with lower concentrations was still far away. The influence distance reaches 3.28 km at the end of 20 years, endangering the water supply security of the Xixiazhuang area. Therefore, after taking strong pumping measures in the four Hougao wells, there was still a wide range of contaminant exceeding the standard, and further control measures are needed to control the spread of pollution exceeding standard.

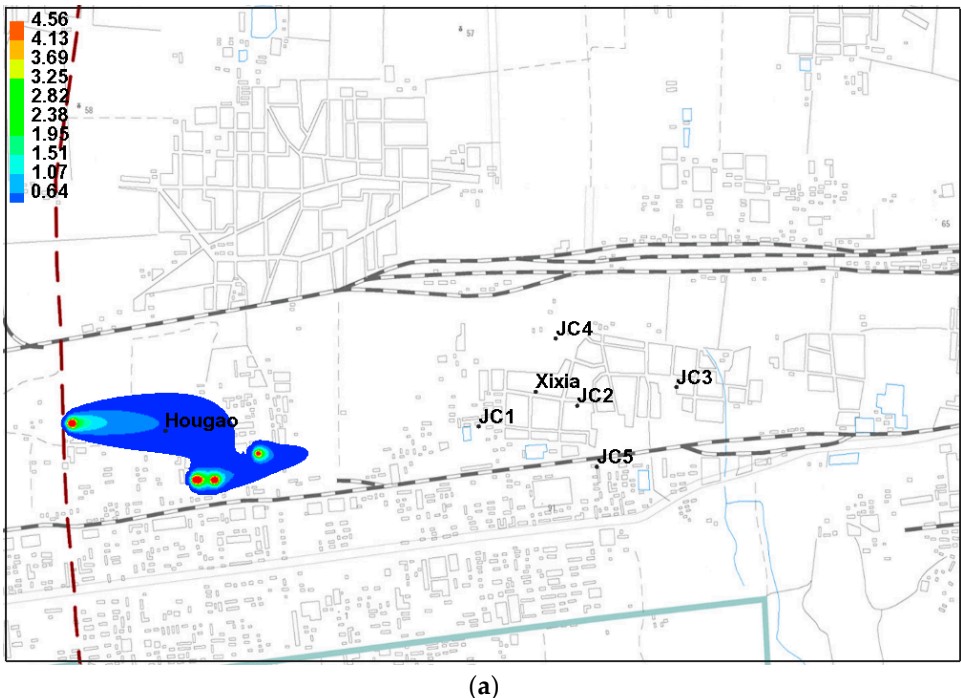

(**a**)

**Figure 12.** *Cont.*

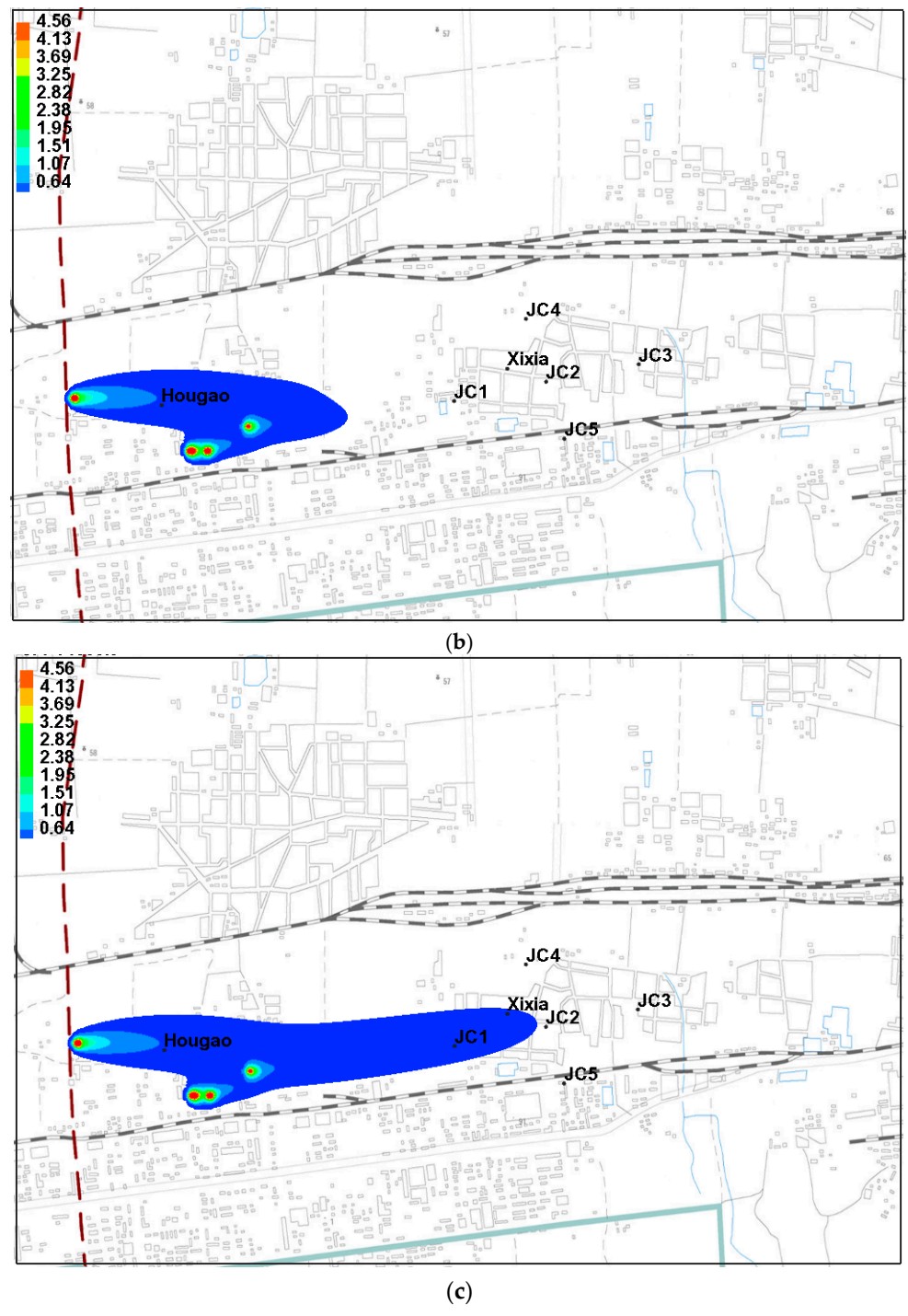

**Figure 12.** Prediction diagram of ammonia nitrogen transport in the four Hougao wells with strong pumping; (**a**) five years; (**b**) ten years; (**c**) twenty years.

According to the monitoring data of the monitoring well, the concentration changing curve is drawn as shown in Figure 13. It can be seen that the peak concentration of the monitored well decreases significantly, and the maximum peak value decreases from 2.32 mg/L to 0.58 mg/L when no measures are taken, and only the concentration slightly exceeds the standard. It shows that the strong pumping wells have a certain effect on pollutant control, but it does not achieve the effect of pollution control completely. The contaminant concentration of monitoring wells JC1 and JC2 was still slightly higher than the limit of the groundwater standard class III.

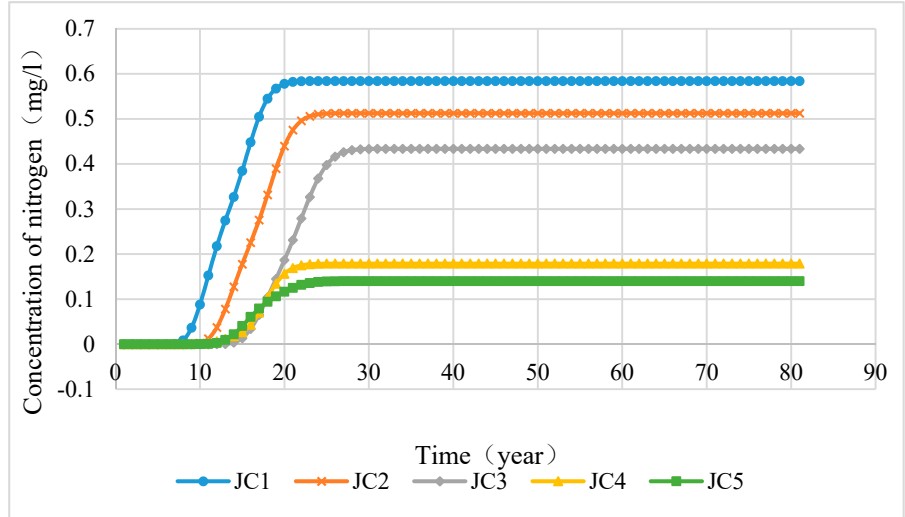

**Figure 13.** Ammonia nitrogen concentration changing curves of five monitoring wells with strong pumping.

*4.4. Control Effect Analysis of Hydraulic Interception Wells Set to the East of the Four Hougao Wells on the Pollutant*

Since the strong pumping well was still unable to completely control the diffusion of the pollutant exceeding the standard, it was considered to set four hydraulic interception wells at the distance of 500 m to the east of the four Hougao wells. A hydraulic interception wall was formed with the effect of strong pumping and drainage of the four hydraulic interception wells, so as to further control the eastward diffusion of the pollutants exceeding the standard. The layout of the interceptor wells is shown in Figure 14. The interceptor wells were pumped with a pumping volume of 5000 m³/d. According to the simulation results shown in Figure 15, the pollutant basically no longer spreads to the eastern region under the control of the hydraulic interception well but is firmly controlled in the vicinity of the interception well. The maximum influence distance of the concentration boundary corresponding to the water standard limit of class III reduces to 1.8 km. The pollutant migration was effectively controlled.

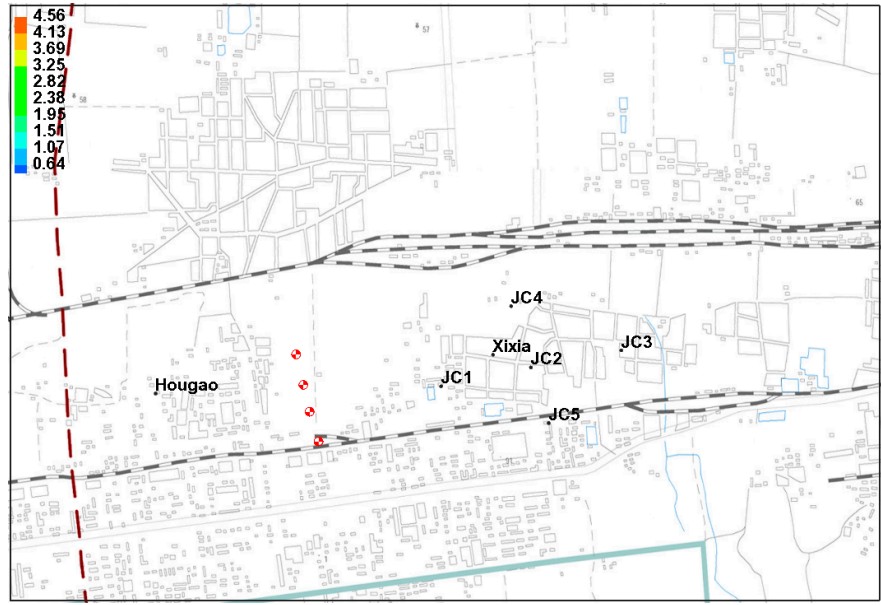

**Figure 14.** The arrangement of hydraulic interception wells.

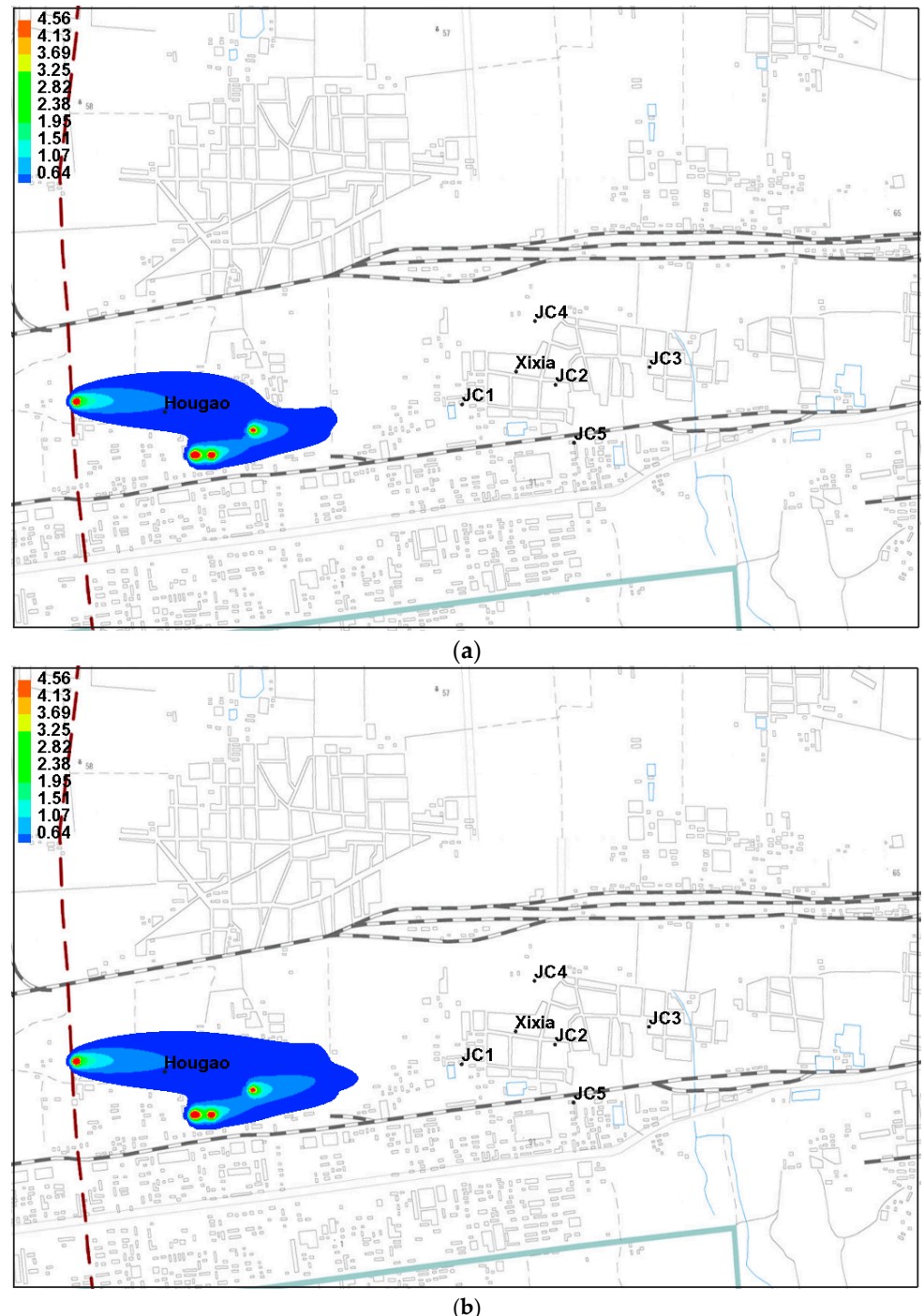

(**a**)

(**b**)

**Figure 15.** *Cont.*

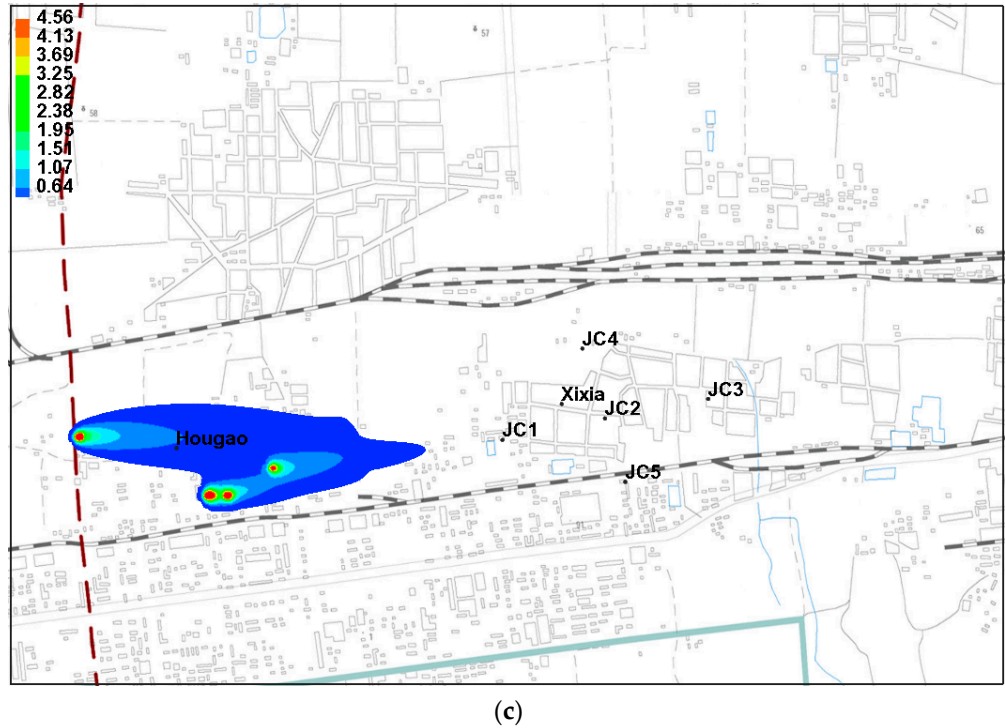

(**c**)

**Figure 15.** Prediction diagram of ammonia nitrogen transport under the combined action of Hougao strong pumping wells and hydraulic interception wells; (**a**) five years; (**b**) ten years; (**c**) twenty years.

The concentration change curve is drawn according to the concentration monitoring data of the monitoring well, as shown in Figure 16. The peak concentration of the monitored well was only 0.11 mg/L. The pollutant concentrations in the Xixia area were all lower than the water standard limit of class III, indicating that the pollutant concentrations were effectively controlled after the adoption of hydraulic interception wells.

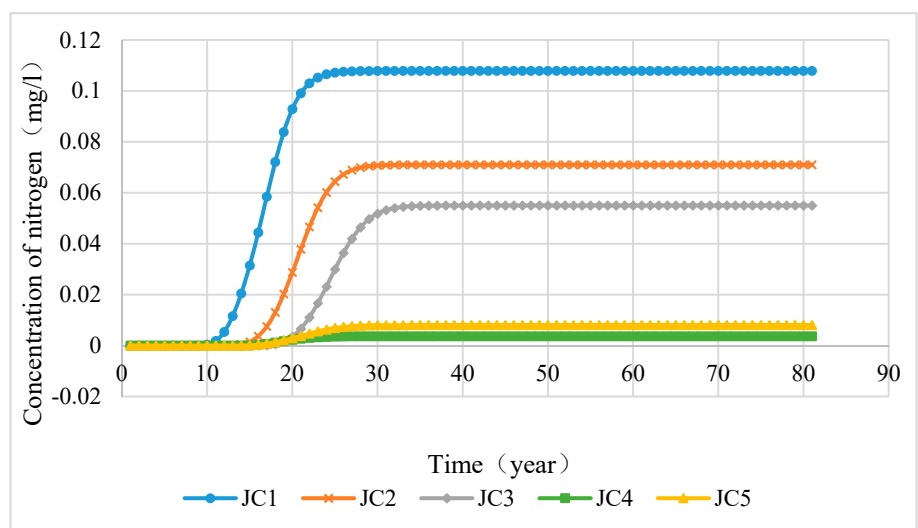

**Figure 16.** Ammonia nitrogen concentration changing curves of five monitoring wells under the joint action of strong pumping wells and hydraulic interception wells.

At the same time, the ammonia nitrogen concentration changing curves of monitoring well JC2 under three working conditions, i.e., no measures are taken, only strong pumping wells, both strong pumping wells and hydraulic interception wells, are shown in Figure 17. It can be found from the

comparison chart that the control effect of adopting prevention and control measures on pollutant concentration is extremely obvious. When the strong pumping well and the hydraulic interception well work together, the pollutant control effect is the best, and the peak pollution concentration is far lower than the standard limit of 0.5 mg/L.

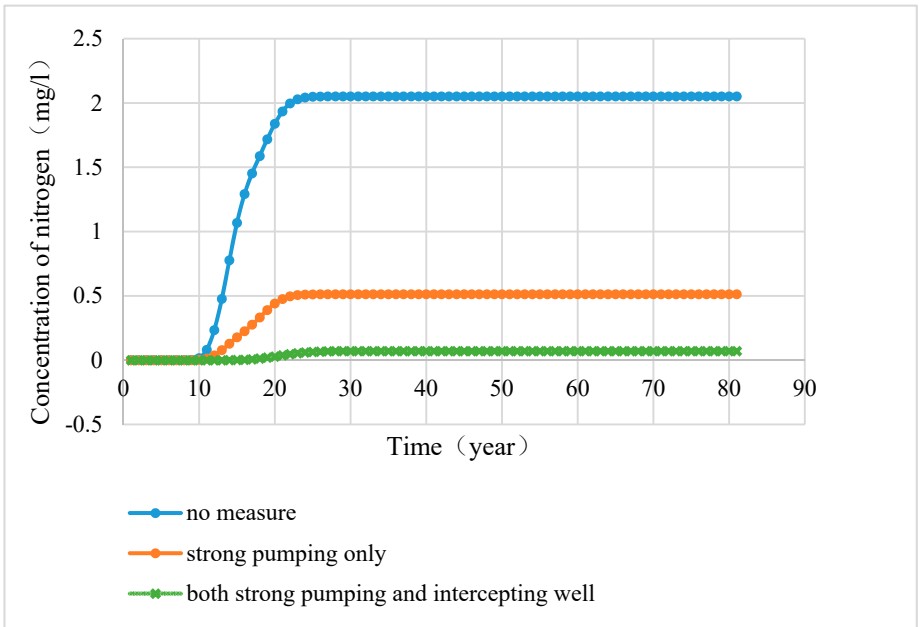

**Figure 17.** Comparison of ammonia nitrogen concentration changing curves in monitoring well JC2 under three working conditions.

## 5. Conclusions

(1) The three-dimensional geological model of the Hougao area of the Dawu water source area was established by analyzing and summarizing the hydrogeological data of the Hougao area of the Dawu water source area. The ammonia nitrogen was taken as an example, and the transport and diffusion of the pollutant in the Hougao area were simulated. It was found that the pollutant was mainly controlled by the convection of groundwater and moves with the flow direction of groundwater to the northeast. In the absence of any control measures, the pollutant will quickly spread to the eastern centralized water supply area, posing a threat to water supply security.

(2) From the concentration change curve of pollution monitoring wells, it can be seen that the closer to the pollution source, the faster the concentration rises and the larger the maximum peak value, and the pollutant concentration decreases from the pollution halo center to both sides.

(3) After taking strong pumping measures in the four Hougao wells, the transport of ammonia nitrogen was effectively controlled, and the peak concentration of monitoring well drops significantly, from 4–5 times of exceeding the standard to slightly exceeding the standard, but there was still a large range of exceeding the standard in Xixia area.

(4) Under the combined action of hydraulic interception wells and strong pumping wells, the scope of ammonia nitrogen was basically controlled in the vicinity of interception wells and does not continue to spread to the eastern water supply area. The maximum monitoring concentration value of pollution was 0.11 mg/L, which is within the groundwater standard limits of class III, and the pollutant control effect is the best, which provides a certain reference for similar pollution control works.

**Author Contributions:** Conceptualization, J.Z.; Methodology, H.Z. and C.J.; Software, S.Y.; Validation, J.W. and L.Y.; Formal Analysis, Z.W.; Investigation, L.Y. and Z.L.; Data Curation, H.L.; Writing-Review & Editing, S.Y. and Z.L.

**Funding:** This research received no external funding.

**Conflicts of Interest:** There is no conflict of interest.

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
