# Peer review of "Control Effects of Hydraulic Interception Wells on Groundwater Pollutant Transport in the Dawu Water Source Area"

_water, doi:10.3390/w11081663_

Round 1

Reviewer 1 Report

L17: Typo. L31: Can you specify the pollutant here? I don't think you meant just one type of contaminant. Figure 1: Picture resolution should be improved, and the legend should be enlarged to some extent, particularly for the inset. The same comment about the figure resolution issue applies for other figures below. The manuscript should be went through and revised by a native English writer.

Author Response

Thank you for taking time out of your busy schedule to review my article carefully. Aiming at your first question:There are many types of groundwater pollutants in the Hougao area. Among them, ammonia nitrogen, chloride, petroleum and benzene exceed the standard most seriously. In order to facilitate the research, we selected one typical pollutant for in-depth study. The ammonia nitrogen is used as the control index of domestic and industrial waste water in the policy documents of pollution emission. It can show the specific situation of industrial waste water and domestic waste water pollution changing with time. So the ammonia nitrogen with high exceeding standard is selected as the pollution factor in this simulation.

Reviewer 2 Report

Authors have improved every remarks. Article is acccepted for me in a present form. 

Author Response

The resolution of the original picture has been improved.

Reviewer 3 Report

Th author updated the manuscript highlighting the importance of the application in the study area and adding some important calibration information. The manuscript is surely enforced but the problem of engliEn remain. The work need a deep English revision and rephrasing. Some periods especially in the introduction are hard to follow. There are also a lot of editing errors.

Author Response

We had improved the wording and structure of the manuscript especially the part of introduction.

Round 2

Reviewer 1 Report

The authors have addressed my comments and suggestions. I think the revision is ready for publication.

Reviewer 3 Report

I don't see many differences between this version and the oldest one. I suggest more attention mainly in the part highlighted in the pdf version i provided.

This manuscript is a resubmission of an earlier submission. The following is a list of the peer review reports and author responses from that submission.

Round 1

Reviewer 1 Report

Introduction: The scientific assumption and rationale for this study were not clearly stated. The authors just listed some refs but did not provide scientific in-depth discussion on how the MODFLOW was used for these studies. Also, a bigger picture about the status and significance of Dawu water source in Northern China should be clearly emphasized. I think the whole introduction section should be rewritten and improved significantly. Section 2.2: Provide relevant refs to support the statements here. Figure 1: Any figure captions or descriptions for different colorful regions shown in the figure? Section 3.2: Very limited information was provided about how to build the model and select the parameters. This section should be strengthened to some extent. Table 1: Can you explain how to generate the hydrogeological parameters for the model shown in the table? Detailed information is needed for how to do the model calibration and validation. This is very important but unfortunately it is missing here and also in the Results and Discussion section. Section 4.1: It is so weird to talk about the pollutant here since no information has been given for pollutant before. I think an introduction for pollutants examined here should be given either in the Introduction or the M&M section. And the same thing for the pumping (section 4.2), please do as suggested shown above. Section 4.1: I think the authors should also state clearly how did you collect the data for ammonia, nitrogen, chloride, and other contaminants used here. The manuscript should be double checked by a native English speaker in terms of for example grammatical issue, improper use of English, and logistic flow. Based on the above comments and concerns, I do not think the manuscript can be accepted for publication by the journal.

Reviewer 2 Report

There are some comments to the article:

- abstract should be rearranged - please add some part with conclusions

-please highlight the purpose of the article in the introduction

there is no introduction to the used methods

-please describe in detail the relationship between monitoring and modeling (what data were used, why)

-what is the accuracy of the model?

-please indicate results of the research

-the section devoted to the conclusions should be more concentred on the results

-improve English

Reviewer 3 Report

The paper aim to the development of a MODFLOW project for pollution transport. The paper look more like a technical report and this represent a big problem. Some important figure like study are and lithological profile are completely missing and the figure presented are not self-contained. I think a full English revision and rephrasing is mandatory like a complete reorganization of article structure